# The Role of [^18^F]F-Choline PET/CT in the Initial Management and Outcome Prediction of Prostate Cancer: A Real-World Experience from a Multidisciplinary Approach

**DOI:** 10.3390/biomedicines10102463

**Published:** 2022-10-01

**Authors:** Luca Urso, Giovanni Christian Rocca, Francesca Borgia, Federica Lancia, Antonio Malorgio, Mauro Gagliano, Mauro Zanetto, Licia Uccelli, Corrado Cittanti, Carmelo Ippolito, Laura Evangelista, Mirco Bartolomei

**Affiliations:** 1Department of Translational Medicine, University of Ferrara, Via Aldo Moro 8, 44124 Ferrara, Italy; 2Nuclear Medicine Unit, Oncological Medical and Specialist Department, University Hospital of Ferrara, 44124 Ferrara, Italy; 3Urology Unit, Surgical Department, University Hospital of Ferrara, 44124 Ferrara, Italy; 4Oncology Unit, Oncological Medical and Specialists Department, University Hospital of Ferrara, 44124 Ferrara, Italy; 5Radiotherapy Unit, University Hospital of Ferrara, 44124 Ferrara, Italy; 6Hospital Radiology, University Hospital of Ferrara, 44124 Ferrara, Italy; 7Department of Medicine DIMED, University of Padua, 35128 Padua, Italy

**Keywords:** [^18^F]F-choline PET/CT, prostate cancer, conventional imaging, outcome

## Abstract

Initial staging of prostate cancer (PCa) is usually performed with conventional imaging (CI), involving computed tomography (CT) and bone scanning (BS). The aim of this study was to analyze the role of [^18^F]F-choline positron emission tomography (PET)/CT in the initial management and outcome prediction of PCa patients by analyzing data from a multidisciplinary approach. We retrospectively analyzed 82 patients who were discussed by the uro-oncology board of the University Hospital of Ferrara for primary staging newly diagnosed PCa (median age 72 (56–86) years; median baseline prostate specific antigen (PSA) equal to 8.73 ng/mL). Patients were divided into three groups based on the imaging performed: group A = only CI; group B = CI + [^18^F]F-choline PET/CT; group C = only [^18^F]F-choline PET/CT. All data on imaging findings, therapy decisions and patient outcomes were retrieved from hospital information systems. Moreover, we performed a sub-analysis of semiquantitative parameters extracted from [^18^F]F-choline PET/CT to search any correlation with patient outcomes. The number of patients included in each group was 35, 35 and 12, respectively. Patients with higher values of initial PSA were subjected to CI + PET/CT (*p* = 0.005). Moreover, the use of [^18^F]F-choline PET/CT was more frequent in patients with higher Gleason score (GS) or ISUP grade (*p* = 0.013). The type of treatment performed (surgery *n* = 33; radiation therapy *n* = 22; surveillance *n* = 6; multimodality therapy *n* = 6; systemic therapy *n* = 13; not available *n* = 2) did not show any relationship with the modality adopted to stage the disease. [^18^F]F-choline PET/CT induced a change of planned therapy in 5/35 patients in group B (14.3%). Moreover, patients investigated with [^18^F]F-choline PET/CT alone demonstrated longer biochemical recurrence (BCR)-free survival (30.8 months) in comparison to patients of groups A and B (15.5 and 23.5 months, respectively, *p* = 0.006), probably due to a more accurate selection of primary treatment. Finally, total lesion choline kinase activity (TLCKA) of the primary lesion, calculated by multiplying metabolic tumor volume and mean standardized uptake value (SUVmean), was able to more effectively discriminate patients who had recurrence after therapy compared to those without (*p* = 0.03). In our real-world experience [^18^F]F-choline PET/CT as a tool for the initial management of PCa had a relevant impact in terms of therapy selection and was associated with longer BCR-free survival. Moreover, TLCKA of the primary lesion looks a promising parameter for predicting recurrence after curative therapy.

## 1. Introduction

Prostate cancer (PCa) is the most common malignancy in males, with an incidence of approximately 1.4 million cases per year worldwide [1,2]. Indeed, accurate detection of disease extension (prostatic vs. extra-prostatic) is a key factor in treatment approach selection [3]. According to the most recent guidelines published by the European Association of Urology (EAU) and the European Association of Nuclear Medicine (EANM) [4], the use of imaging modalities for disease staging is mandatory after primary diagnosis of clinically relevant PCa (International Society of Urological Pathology (ISUP) grade ≥ 2). However, the best staging scheme is still a matter of debate [5]. Conventional imaging (CI) techniques used for staging PCa are magnetic resonance (MR), computed tomography (CT) and bone scanning (BS), but they all have limitations. MR, in particular with the multiparametric protocol (MRmp), represents the gold standard imaging modality for T-staging, but its acquisition window is limited to the pelvic region, enabling extra-pelvic metastases to be investigated. CT has very low diagnostic value in T-staging, as it lacks specificity for distinguishing benign from malignant forms [6]. Moreover, both CT and MR present low sensibility in distinguishing metastatic from non-metastatic lymph nodes as the only discrimination criteria is based on their size and morphology [4,7]. As for M-staging, BS outperforms CT in detecting skeletal spread of PCa but has a limited detection rate in the case of small lesions and lacks specificity, leading to some doubtful results [8,9]. Therefore, in recent years, positron emission tomography (PET) associated with CT or MRI (PET/CT or PET/MRI) has been increasingly utilized in PCa staging [10,11]. PET imaging provides a functional evaluation to support and integrate the morphologic study. Moreover, it has the potential to provide a “one stop shop” examination, able to perform T-, N- and M-staging in a single acquisition [10]. Most recent guidelines suggest that staging PET/CT or PET/MRI should be performed as second-line imaging in the case of doubtful cases for CI or as first-line imaging in patients with high-risk PCa (i.e., clinical T3-T4, Gleason score (GS) higher than 7 and prostate specific antigen (PSA) higher than 10 ng/mL) [4].

Several PET radiotracers are available for PCa imaging. The first radiopharmaceutical employed for staging and restaging PCa was choline, radiolabeled with ^11^C or ^18^F [12]. Choline is a component of phosphatidylcholine, one of the main constituents of cell membranes [13]. Several studies have shown upregulation of choline kinase in malignant transformation of cancer cells, particularly PCa [14]. To date, besides staging indications described above, choline PET is utilized for detecting recurrent PCa in the case of PSA elevation during follow-up [15]. More recently, new radiopharmaceutical agents both for staging and restaging PCa were introduced, such as radiolabeled ligands of prostate-specific membrane antigen (PSMA) [16,17]. [^68^Ga]Ga-PSMA-11, the most utilized, allows the detection of lesions expressing PSMA, which is directly related to the GS and to the aggressive behavior of PCa [18]. Thus, PSMA imaging in PCa contributes to distinguishing low-risk from high-risk cancers [18].

Although PSMA PET has been shown to be more sensitive and more specific than choline PET in the identification of biochemical recurrence of PCa, especially in the post-RP setting, providing salvage-treatment guidance and predicting clinical outcomes [19,20], the literature currently lacks direct comparisons between the two radiotracers in the staging setting [21]. Furthermore, PSMA PET is only performed in a relatively small number of diagnostic centers compared with choline PET. As a consequence, radiolabeled choline remains the most widespread radiotracer for PET imaging.

The aim of this study was to analyze the role of [^18^F]F-choline PET/CT in the initial management and outcome prediction of PCa patients by analyzing data from a multidisciplinary approach. As a secondary aim, we evaluated whether semi-quantitative parameters, extracted from [^18^F]F-choline PET/CT, could discriminate between patients who had recurrence after primary treatment.

## 2. Materials and Methods

### 2.1. Patient Selection

We retrospectively analyzed 82 patients referring to the multidisciplinary uro-oncologic board of the University Hospital of Ferrara for primary staging newly diagnosed PCa. Patients selected were discussed during a period of 46 months, from April 2018 to February 2022. For each patient, a wide range of clinical–pathological data were retrieved from the hospital information systems, including body mass index (BMI), familiarity for PCa, PSA levels at initial diagnosis, GS, ISUP grade, prostate biopsy, MRmp results, staging strategy, type of therapy selected and outcomes (in terms of overall survival (OS), progression-free survival (PFS) and biochemical recurrence (BCR)).

Patients were divided into three subgroups based on the staging strategy decision: group A (patients who underwent staging with CI, involving thorax abdomen contrast enhanced CT and/or BS); group B (patients who underwent both CI and [^18^F]F-choline PET/CT); and group C (patients who only underwent [^18^F]F-choline PET/CT staging). For patients in groups A and B, the different diagnostic investigations were performed within a month. In group B patients, [^18^F]F-choline PET/CT was usually performed as a second-line exam after CI to investigate inconclusive findings. Conversely, [^18^F]F-Choline PET/CT alone was proposed by the multidisciplinary team in the case of high risk of metastasis at diagnosis (based on Gleason score and/or high PSA value and/or presence of risk findings at mpMR). The number of patients included in groups A, B and C was 35, 35 and 12, respectively. We analyzed the patients’ distribution into the 3 groups in relation to baseline clinical pathological characteristics. Moreover, we investigated whether the different staging strategies were correlated with therapy decision and outcome. Finally, for patients included in group B, we analyzed whether [^18^F]F-choline PET/CT induced a change into multidisciplinary therapeutic decision in comparison to CI findings.

### 2.2. CI Acquisition and Interpretation

CT was performed with and without contrast enhancement administration on a Brilliance iCT 128 SP scanner (Philips Healthcare, Amsterdam, The Netherlands). Imaging analysis was performed by an expert radiologist with more than 10 years of experience in uro-oncologic disease.

BS was performed on a Symbia T2 dual-head hybrid SPECT/CT system (Siemens Healthineers, Erlangen, Germany), 3 h after intravenous injection of 740–925 MBq of [99 mTc]Tc-hydroxydiphosphonate. The patient was required to hydrate with at least 1 L of water between the radiotracer injection and the diagnostic scan. BS images were evaluated by an expert nuclear medicine physician, with more than 20 years of experience.

### 2.3. [^18^F]F-Choline PET/CT Acquisition and Interpretation

Images were acquired from the mid-thigh to the skull vertex about 60 min after [^18^F]F-choline injection (3 MBq/Kg) using a standard technique on a dedicated PET/CT system (Biograph mCT Flow Motion; Siemens Medical Solutions, Malvern, PA, USA). After non-contrast-enhanced low-dose CT (120 keV, 80 mAs, CareDose; reconstructed with a soft-tissue kernel to a slice thickness of 3 mm), PET was acquired in 3-dimensional mode (matrix, 200 × 200) using a continuous scan with variable speed. The emission data were corrected for randoms and scatter, and decay and attenuation correction was performed using the non-enhanced low-dose CT data. Reconstruction was performed on a Syngo.via workstation (Siemens Healthineers, Erlangen, Germany). All images were processed and analyzed by two experienced nuclear medicine physicians. Circular regions of interest (ROIs) were drawn around the tumor lesions with focally increased uptake in transaxial slices and automatically adapted to a 3-dimensional the volume of interest (VOI) (Figure 1). The system automatically calculated standardized uptake value (SUV)max, SUVmean and two volumetric parameters: metabolic tumor volume (MTV), which represents the tumor volume with at least 40% uptake of the SUVmax within the VOI, and total lesion choline kinase activity (TLCKA), which is calculated multiplying SUVmean and MTV within the same VOI. Quantitative parameters were calculated for the primary tumor. Moreover, in the metastatic disease, MTV and TLCKA were reported for the reference lesion (most relevant lymph node and bone metastasis in terms of extension and uptake intensity) and for the whole body load of disease, both for N- (MTV_∑N and TLCKA_∑N) and M-staging (MTW_∑M and TLCKA_∑M). Finally, the sum of MTV and TLCKA values, referring to every lesion detected, was calculated (MTV_WB and TLCKA_WB).

### 2.4. Outcome

As the outcome parameter we used biochemical recurrence (BCR), considered as the time between the prostate biopsy and the time of evidence of increased levels of PSA after primary treatment (PSA > 0.20 ng/mL in patients treated with surgery; PSA increase of at least 2 ng/mL in patients treated with radiation therapy or systemic therapy). Data were acquired from the hospital’s digital archive.

### 2.5. Statistical Analysis

Categorical and continuous data were expressed as percentage and median (range) or as mean ± standard deviation (SD) as appropriate. For the comparison among categorical data, the chi-squared test was used, while for the continuous data, parametric or non-parametric tests were applied. The statistical significance threshold was set as a *p* value less than 0.05. Statistical analysis was performed by using MedCalc version 20.027 (MedCalc Software Ltd, Ostend, Belgium).

## 3. Results

Overall, 82 patients were enrolled in the study. Patients’ characteristics are summarized in Table 1. Median PSA value was 8.73 ng/mL at the moment of prostate biopsy. Twenty patients underwent random prostate biopsy (24.4%), while in the remaining 62 (75.6%) an MR-guided approach was performed. Gleason score (GS) was <7, =7 and >7 in 15, 40 and 24 patients respectively. GS was not assessed in three patients due to interference with androgen deprivation therapy (ADT).

CT was performed in 52 patients (63.4%), and metastases was detected in 10 (19.2%) subjects. In most cases, metastatic foci were found in pelvic lymph nodes (*n* = 8 patients). In the remaining two patients, one presented only bone metastases and the other both lymph node and bone metastases.

Sixty patients (73.2%) underwent BS, and skeletal metastasis was detected in four subjects (6.7%). In five cases, second-level imaging was requested after BS to further investigate doubtful findings.

[^18^F]F-choline PET/CT was performed in 47 patients (57.3%), and every scan detected at least one pathological uptake indicative of PCa localization. In particular, in 28 cases (59.6%) [^18^F]F-choline PET/CT detected only primary tumor, while in 17 patients (36.2%) a pathological uptake was detected for prostate and lymph node (median number of lymph node metastases were equal to 3) and/or bone metastases (median number of bone metastases was equal to 2). In total, PET detected 56 metastatic lesions (*n* = 15 bone lesions and *n*= 41 metastatic lymph nodes). In two patients (4.3%) [^18^F]F-choline PET/CT detected only bone metastases, without significant uptake in the primary tumor. Finally, no patients showed visceral metastases. In [^18^F]F-choline PET/CT quantitative analysis, median MTV_WB and TLCKA_WB were 10.58 and 54.14, respectively.

Overall, the patients were subjected to several different therapies, described in Table 2. Overall, 13 patients (15.9%) had BCR after primary treatment, at a median time of 16.2 months.

Results of a sub analysis, where patients were divided into three groups based on the staging strategy, are displayed in Table 3. Patients with higher values of initial PSA were more frequently included in group B (*p* = 0.005). Moreover, the use of [^18^F]F-choline PET/CT was more frequent in patients with higher GS or ISUP grade (*p* = 0.013). The type of treatment performed did not show any relationship with the modality staging strategy. However, a slightly different treatment approach was used for the three groups. Patients from group A were more often treated by local therapies (surgery and RT) or active surveillance than groups B and C. Furthermore, group C patients were often subjected to systemic therapy. [^18^F]F-choline PET/CT induced a change of planned therapy in 5/35 patients in group B (14.3%). In particular, [^18^F]F-choline PET/CT upstaged four patients that advanced from radical treatment to systemic therapy due to metastatic disease. In the remaining patient, [^18^F]F-choline PET/CT did not show any uptake in skeletal lesions detected by CT. As a consequence, the patient underwent radical radiation therapy on the prostate bed that induced a fall in PSA values (from 3.64 ng/mL to 0.18 ng/mL). Notably, the patient had still not been incurred in BCR after the 26-month follow-up, and the skeletal lesions had not been modified on follow-up CT images. Moreover, patients who underwent only [^18^F]F-choline PET/CT demonstrated longer BCR-free survival (30.8 months) in comparison to patients of groups A and B (15.5 and 23.5 months, respectively *p* = 0.006).

The analysis of semiquantitative parameters extracted from [^18^F]F-choline PET/CT (Table 4) demonstrated that only TLCKA of the primary lesion was able to discriminate patients who had recurrence after therapy (*p* = 0.03).

## 4. Discussion

Correct primary staging is of great importance for offering the best therapeutic management to every newly diagnosed PCa patient. Much of the evidence in the literature agrees on the better accuracy of [^18^F]F-choline PET/CT over CI in primary staging of intermediate to high-risk PCa [3,5,22,23,24,25]. Beheshti et al. [3] reported that [^18^F]F-choline PET/CT was useful for excluding extra-prostatic disease spread in a prospective cohort of 132 patients (66% sensitivity, 96% specificity, 82% positive and 92% negative predictive value). Their results are consistent with those published by Poulsen et al. [24] and Metser et al. [22]. Similarly, Evangelista et al. [5] found that [^18^F]F-choline PET/CT outperformed CI for N-staging (sensitivity 69.2 vs. 46.2, respectively). However, guidelines [4] still recommend [^18^F]F-choline PET/CT in a limited number of cases, mainly on the basis of other studies whose results are in contrast with those listed above [26,27,28]. Therefore, clinicians often debate over patients’ imaging management, and this debate has been furthered by the advent of PSMA-ligand PET/CT, which demonstrated very high performance in this subset of patients, which was probably even better than that obtained by [^18^F]F-choline PET/CT, even though a direct comparison of the two radiotracers in the primary staging setting is yet to be published [29,30,31]. Beyond theory, with this paper we provide a real-world experience regarding the primary staging planning of PCa patients discussed in our institutional multidisciplinary uro-oncologic board. From our data, it can be seen that [^18^F]F-choline PET/CT staging was most frequently used in the case of GS > 7 or inconclusive CI results (*p* = 0.013), in line with current guidelines. Moreover, patients with higher values of baseline PSA underwent a higher number of imaging examinations (*p* = 0.005). Furthermore, we found that patients who underwent only [^18^F]F-choline PET/CT (group C) showed significantly longer BCR-free survival (30.8 months) in comparison to groups A and B (*p* = 0.006), even though their initial risk was higher. Moreover, contrary to our expectations, patients in group A had shorter BCR-free survival in comparison to those for whom [^18^F]F-choline PET/CT was associated with CI (15.5 vs. 23.5 months), despite starting with a lower PCa class of risk (lower GS and lower mean baseline PSA values). To the best of our knowledge, this is the first paper reporting longer BCR-free survival in patients staged with [^18^F]F-choline PET/CT in comparison to those staged with CI. We believe that this result may reflect the better accuracy offered by [^18^F]F-choline PET/CT staging in our cohort of patients, which induced a better treatment selection and, consequently, a benefit in terms of outcome. Moreover, the evidence that group C patients showed longer BCR-free survival if compared to patients in group B suggests that CI does not provide an added value in PCa patients undergoing primary staging with [^18^F]F-choline PET/CT. If confirmed in larger studies, we could even speculate that CI represents a confounding factor in some subset of patients undergoing primary staging with 18F-choline PET/CT, inducing a subsequent inappropriate therapeutic decision. [^18^F]F-choline PET/CT induced a change in treatment selection in 14.3% of patients of group B, in line with previous results in the literature [3,5,24]. In particular, the uptake corresponding to small lymph nodes, below significance for CT dimensional criteria and the early detection of small bone metastasis without skeletal modifications in CI determined an increased stage in four patients that consequently underwent systemic therapy (Figure 2). Moreover, [^18^F]F-choline PET/CT ruled out suspected bone metastasis in another patient, allowing successful prostate radiation therapy.

Another advantage of [^18^F]F-choline PET/CT staging is that it provides “one stop shop” imaging, as it allows a metabolic evaluation of the prostate, beside N- and M-staging. However, it is described in the literature that several pathological conditions can induce an increased prostate tracer uptake, both benign and malignant [11]. In our experience, [^18^F]F-choline PET/CT showed an increased prostate uptake in 45/47 patients (95.7%). Nevertheless, the prostate pattern of uptake was very inhomogeneous, as it was focal and intense in some patients and mild and diffuse in some others. Laudicella et al. [32] recently reported that infiltrative growth pattern is related to a significantly lower uptake for [^68^Ga]Ga-PSMA-PET/MRI compared to the expansive growth pattern. A similar correlation should also be investigated for [^18^F]F-choline PET/CT or possibly [^18^F]F-choline PET/MRI. Moreover, radiomics and artificial intelligence have been rarely applied to [^18^F]F-choline PET/CT [33]. Future work to identify PET features able to help the nuclear medicine physician in the T-staging evaluation should be considered.

Volumetric parameters (VPs) extracted by PET images are promising and widespread diagnostic and prognostic tools in several types of cancers and for many PET radiotracers [34,35]. However, few papers have explored their potential applications in [^18^F]F-choline PET/CT. Tseng et al. [36] reported that TLCKA of primary tumors (that the authors named uptake volume product, UVP) was a significant predictor for PFS, together with some MR-derived parameters. Moreover, Kim et al. [37] found a correlation between TLCKA (named UVPMRI) and PCa characteristics (GS, tumor volume and baseline PSA). We found that TLCKA of the primary lesion was the only VP able to predict biochemical recurrence. This is consistent with the results published by Sepulcri et al. [38] who reported that TLCKA of primary tumors in [^18^F]F-choline PET/CT staging was a predictor of BCR after radiation therapy. TLCKA of the primary tumor reflects both the local extension of PCa as well as its choline metabolism. Interestingly, TLCKA_WB could not predict BCR. This means that in our cohort, the PET-characteristics of the primary tumor were more important than the PET whole body load of disease for discriminating patients who would experience BR.

Some limitations should be considered when analyzing the results of this paper. Indeed, this study is retrospective, with a medium-sized sample. Prospective trials with larger cohort of patients to confirm our data would be desirable. Moreover, it was not possible to calculate the change of therapeutic management in patients in group C, due to the absence of comparative CI data available.

## 5. Conclusions

This real world-experience showed that primary staging of PCa patients with [^18^F]F-choline PET/CT was associated with longer BCR-free survival in comparison to CI. Moreover, we identified a prognostic factor in TLCKA of the primary lesion that was able to discriminate patients who would have recurrence after primary treatment. We are strongly convinced that multidisciplinary discussion is an added value for selecting the best primary staging scheme for every newly diagnosed PCa patient. Despite the rise of PSMA-ligand PET imaging, our real-world experience suggests that [^18^F]F-choline PET/CT should still be considered a valid and reliable diagnostic tool for primary staging of high-risk PCa patients or, in the case of inconclusive CI, providing an outcome improvement.

## Figures and Tables

**Figure 1 biomedicines-10-02463-f001:**
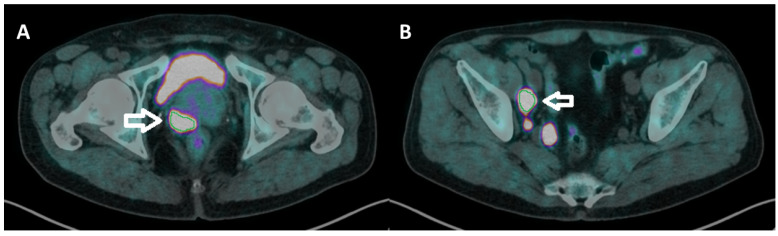
(**A**,**B** transaxial fused images) manual workflow of [^18^F]F-choline PET/CT imaging elaboration. A circular ROI was drawn on prostate cancer ((**A**), white arrow) and lymph node metastasis ((**B**), white arrow) and automatically adapted into a 3-dimensional VOI, from which the software directly extracted semiquantitative and volumetric (threshold = 40%) parameters.

**Figure 2 biomedicines-10-02463-f002:**
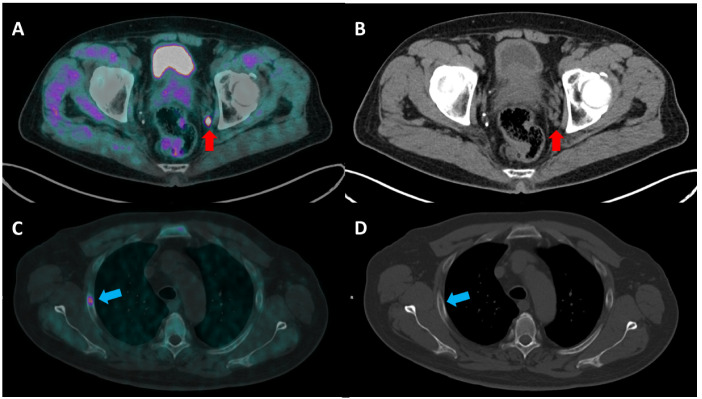
(**A**,**B**) A 71-year-old man with locally advanced PCa (Gleason score 9, 4 + 5 and a serum PSA value of 36 ng/mL) underwent [^18^F]F-choline PET/CT ((**A**) transaxial fused image; (**B**) CT image) for the initial staging of the disease. The PET/CT image shows a pathological uptake corresponding to a small left internal iliac lymph node (red arrow). Lymph node metastasis was not detected in CT staging, as its size (0.7 × 0.6 mm) was below the significance criteria. (**C**,**D**) A 68 year-old-man with PCa (G.S 6, 3 + 3; PSA value of 71.29 ng/mL) was imaged with [^18^F]F-choline PET/CT staging ((**C**) transaxial fused image; (**D**) CT image), with evidence of a focal uptake corresponding to a bone lesion at the third right rib (blue arrow), which could not be detected by CT due to the normal bone structure.

**Table 1 biomedicines-10-02463-t001:** Patients’ characteristics and results of diagnostic staging exams.

Variables	
Patients enrolled, number	82
Median age (range), years	72 (56–86)
Median BMI (range)	27 (22–36)
Rectal examination Negative Positive Doubt NA	30 (36.6%) 17 (20.7%) 5 (6.1%) 30 (36.6%)
Familiarity No Yes NA	33 (40.2%) 5 (6.1%) 44 (46.3%)
Median baseline PSA (range), in ng/mL	8.73 (1.81–56.5)
MRmp imaging No Yes	20 (24.4%) 62 (75.6%)
MRmp results Negative Positive	0 62
Gleason score <7 =7 >7 NA	15 (18.3%) 40 (48.8%) 24 (29.3%) 3 (3.7%)
ISUP 1 2 3 4 5 NA	14 (17.1%) 21 (25.6%) 19 (23.2%) 17 (20.7%) 8 (9.8%) 3 (4.9%)
CT imaging No Yes	30 (36.6%) 52 (63.4%)
CT imaging results Negative Positive for N Positive for M Positive for NM NA	42 (51.2%) 8 (8%) 1 (1.2%) 1 (1.2%) 30 (36.6%)
BS imaging No Yes	22 (26.8%) 60 (73.2%)
BS imaging results Negative Positive Doubt	50 (61%) 4 (4.9%) 5 (6.1%)
PET imaging No Yes	35 (42.7%) 47 (57.3%)
PET imaging results Positive on T Positive on M Positive on TN Positive on TNM	28 (59.6%) 2 (4.3%) 11 (23.4%) 6 (12.8%)
Median MTV_WB (range)	10.58 (2.04–279.79)
Median TLCKA_WB (range)	54.14 (7.58–8677)

BMI: body mass index; BS: bone scanning; CT: computed tomography; ISUP: International Society of Urological Pathology; MRmp: multiparametric magnetic resonance; MTV_WB: whole body metabolic tumor volume; NA: not available; PET: positron emission tomography; PSA: prostate specific antigen; TLCKA_WB: whole body total lesion choline kinase activity.

**Table 2 biomedicines-10-02463-t002:** Patients’ therapy selection and outcomes.

Variables	
Type of therapy Surgery RT Surveillance Multimodality therapy Systemic therapy NA	33 (40.2%) 22 (26.8%) 6 (7.3%) 6 (7.3%) 13 (15.9%) 2 (2.4%)
Change of therapy after PET imaging (only for group B) No Yes	30 (85.7%) 5 (14.3%)
Biochemical recurrence No Yes NA	67 (81.7%) 13 (15.9%) 2 (2.4%)
Median time of biochemical recurrence (range), month	16.23 (1.3–51.87)

NA: not available; PET: positron emission tomography; RT: radiation therapy.

**Table 3 biomedicines-10-02463-t003:** Clinical characteristics, imaging findings and outcome for different groups of patients.

Variables	Group A	Group B	Group C	*p* Value
	Only CI (*n* = 35, 42.7%)	CI + PET (*n* = 35, 42.7%)	Only PET (*n* = 12, 14.6%)	
Mean age (±SD)	70 ± 7	73 ± 6	69 ± 7	0.137
Familiarity No Yes NA	19 (54.3%) 4 (11.4%) 12 (34.3%)	10 (28.6%) 1 (2.8%) 24 (68.6%)	4 (33.3%) 0 8 (66.7%)	0.569
Mean baseline PSA (±SD), in ng/mL	8.2 ± 4.9	15.9 ± 13.9	8.9 ± 5.1	0.005
Gleason score <7 =7 >7 NA	9 (25.7%) 21 (60%) 3 (8.6%) 2 (5.7%)	4 (11.4%) 13 (37.1%) 16 (45.7%) 2 (5.7%)	2 (16.7%) 6 (50%) 4 (33.3%) 0	0.013
ISUP 1 2 3 4 5 NA	9 (25.7%) 14 (40%) 7 (20%) 2 (5.7%) 1 (2.8%) 2 (5.7%)	4 (11.4%) 5 (14.3%) 8 (22.9%) 13 (37.1%) 4 (11.4%) 1 (2.8%)	1 (8.3%) 2 (16.7%) 4 (33.3%) 2 (16.7%) 3 (25%) 0	0.013
Type of therapy Surgery RT Surveillance Multimodality therapy Systemic therapy NA	14 (40%) 9 (25.7%) 3 (8.6%) 3 (8.6%) 5 (14.3%) 1 (2.8%)	14 (40%) 11 (31.4%) 2 (5.7%) 2 (5.7%) 5 (14.3%) 1 (2.8%)	5 (41.7%) 2 (16.7%) 1 (8.3%) 1 (8.3%) 3 (25%) 0	0.984
Recurrence No Yes NA	31 (88.6%) 4 (11.4%) 0	26 (74.3%) 8 (22.9%) 1 (2.8%)	10 (83.4%) 1 (8.3%) 1 (8.3%)	0.311
Mean BCR-free survival (± SD)	15.5 ± 7.5	23.5 ± 12.2	30.8 ± 17.8	0.006

BCR: biochemical relapse; CI: conventional imaging; ISUP: International Society of Urological Pathology; NA: not available; PET: positron emission tomography; PSA: prostate specific antigen; RT: radiation therapy.

**Table 4 biomedicines-10-02463-t004:** Analysis of semiquantitative parameters extracted from [^18^F]F-choline PET/CT.

	No Recurrence (*n* = 36)	Recurrence (*n* = 9)	*p* Value
MTV (primary lesion)	8.75 ± 5.91	12.87 ± 9.54	0.110
TLCKA (primary lesion)	47.05 ± 27.44	74.10 ± 47.31	0.030
SUVmax (primary lesion)	11.1 ± 3.9	12.3 ± 3.4	0.428
MTV_WB	19.1 ± 47.3	23.1 ± 22.6	0.881
TLCKA_WB	323 ± 261	129 ± 51	0.703

## Data Availability

Datasets available upon request.

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
