# Peer review of "The Role of [18F]F-Choline PET/CT in the Initial Management and Outcome Prediction of Prostate Cancer: A Real-World Experience from a Multidisciplinary Approach"

_biomedicines, 2022, doi:10.3390/biomedicines10102463_

Round 1
Reviewer 1 Report
Please find comments in the attachment

Author Response
Thank you for revising our manuscript and for providing useful hints to improve its quality. According to your suggestion, we operated the following changes:
- When it comes to “initial management”, author stated 5/35 of patients were induced a change of planned therapy in group B. Group A (CI) and Group C (PET/CT only) were not discussed. Please discuss group C that when 18F-choline PET/CT is received alone, how many patients resulted in a change of plan.
and
Line 228, “[ 18F]F-choline PET/CT induced a change of planned therapy in 5/35 patients in Group B 228 (14.3%).” Could be also associated with high PSA detected, as indicated in the statement “Patients with higher values of initial PSA were 224 more frequently submitted to Group B (p=0.005).” Therefore, to demonstrate PET indeed helped initial management, change of plans in group C, whom only receives PET while within the cohort of lower PSA should be discussed.
R: based on the suggestions, in material and methods section (line 132), we have better specified the concept of the change of planned therapy. Indeed, we reported only data obtained from comparing patients in the group B (conventional imaging + 18F-Choline PET/CT). As correctly mentioned from the reviewer, in lines 123-127, we added the following sentence: “18F-Choline PET/CT alone was proposed by the multidisciplinary team in case of high risk of metastasis at diagnosis (it means based on Gleason Score and/or high PSA value and/or presence of risk findings at mpMR)”, thus limiting the number of patients who were submitted to a more sophisticated imaging modality. In this latter group of patients (Group C), it was not possible to assess the change of therapeutical management, because no comparative data were available. A small sentence has been added in the limitation paragraph (lines 369-371).
-Statement “Moreover, patients who performed only [18F]F-choline PET/CT demonstrated a longer progression free survival (PFS) (30.8 months) in comparison to patients of Groups A and B (15.5 and 23.5 months, respectively, p=0.006).” could be mis-leading: Diagnostic method could be used to predict PFS, but not, by general understanding, associated with disease stage or treatment. Please re-word.
R: we rephrased the sentence to clarify that we believe that group C had a longer BCR-free survival due to the more accurate primary treatment selection obtained with 18F-choline PET/CT. Therefore, the terms PFS was replaced by BCR-free survival along the manuscript.
- As stated in the line 288-290, please show evidence of higher risk in group C. The only evidence “Patients with higher values of initial PSA were 224 more frequently submitted to Group B (p=0.005).” can be interpreted as group B is the high risk group instead of group C.
and
- “Moreover, patients in Group A had a shorter BCR-free survival in comparison to those who associated [18F] F-choline PET/CT to CI (15.5 vs 23.5 months).” Please explain more on this matter. Group A showed lower GS score population and lower ISUP score, while group B and C showed higher value, this could be contrary to BCR results.
R: thanks for these suggestions. We believe that this is one of the main point of the present manuscript: despite a low class of risk at diagnosis (based on the current EAU guidelines), patients who performed only conventional Imaging had a shorter BCR-free survival, suggesting that they were under-staged and, consequently, undertreated. Indeed, the type of treatment was slightly different among the groups. Patients from Group A were more often treated by local therapies (surgery and RT) or active surveillance than Group B and C, however, Group C were often submitted to systemic therapy. We added this consideration into results section, lines 239-242.
In our opinion, patients of Groups B and C received an appropriate staging and consequently had a better outcome in terms of BCR-free survival. We rephrased lines 305-308, hoping to have clarified this concept for the readers.
- To draw the conclusion/summary in line 295-297, please also explain your hypothesis why group C (PET only) is superior than group B (CI +PET)
R: this is an interesting question. In the lines 123-124, we added that 18F-Choline PET/CT was usually performed after conventional imaging (CI) as a second line investigation to explore inconclusive findings on CI. We believe that the diagnostic flow-chart of primary staging of PCa should be revised, suggesting the use of CI in certain subset of patients and the introduction of new imaging modalities, such as 18F-Choline or radiolabelled PSMA PET/CT or PET/MR in others. However, these assumptions should be confirmed in larger studies, We added this hypothesis in lines 313-318.
Reviewer 2 Report
I would like to congratulate the authors for the topic chosen, how it was enhanced and discussed. The advent of PSMA PET shouldn't lead to the research abandonment of Choline PET, particularly for newly diagnosed cases at primary staging.
However, to improve the manuscript content, the introduction should be enriched (minor revision) by adding a couple of critical references regarding PSMA-PET in order to allow readers to better understand PSMA-PET specifics prior to the methodological development and results explanation of the current Choline-Study:
- "More recently, new radiopharmaceutical agents both for staging and restaging PCa were introduced, such as radiolabeled ligands of prostate-specific membrane antigen (PSMA).”
- Here the following two refs should be inserted:
1)Hofman MS, Lawrentschuk N, Francis RJ, Tang C, Vela I, Thomas P, Rutherford N, Martin JM, Frydenberg M, Shakher R, Wong LM, Taubman K, Ting Lee S, Hsiao E, Roach P, Nottage M, Kirkwood I, Hayne D, Link E, Marusic P, Matera A, Herschtal A, Iravani A, Hicks RJ, Williams S, Murphy DG; proPSMA Study Group Collaborators. Prostate-specific membrane antigen PET-CT in patients with high-risk prostate cancer before curative-intent surgery or radiotherapy (proPSMA): a prospective, randomised, multicentre study. Lancet. 2020 Apr 11;395(10231):1208-1216. doi: 10.1016/S0140-6736(20)30314-7. Epub 2020 Mar 22. PMID: 32209449.
2) Calais J, Ceci F, Eiber M, Hope TA, Hofman MS, Rischpler C, Bach-Gansmo T, Nanni C, Savir-Baruch B, Elashoff D, Grogan T, Dahlbom M, Slavik R, Gartmann J, Nguyen K, Lok V, Jadvar H, Kishan AU, Rettig MB, Reiter RE, Fendler WP, Czernin J. 18F-fluciclovine PET-CT and 68Ga-PSMA-11 PET-CT in patients with early biochemical recurrence after prostatectomy: a prospective, single-centre, single-arm, comparative imaging trial. Lancet Oncol. 2019 Sep;20(9):1286-1294. doi: 10.1016/S1470-2045(19)30415-2. Epub 2019 Jul 30. Erratum in: Lancet Oncol. 2019 Nov;20(11):e613. Erratum in: Lancet Oncol. 2020 Jun;21(6):e304. PMID: 31375469; PMCID: PMC7469487.
- "Despite PSMA PET has been shown to be more sensitive and more specific than Choline PET in the identification of biochemical recurrence of PCa, literature currently lacks of direct comparisons between the 2 radiotracers in staging setting [17]. "
This should be rephrased as follows (plus two additional refs):
" Despite PSMA PET has been shown to be more sensitive and more specific than Choline PET in the identification of biochemical recurrence of PCa, especially in the post-RP setting providing salvage-treatment guidance and predicting clinical outcomes [Ceci et al.; Rovera et al.], literature currently lacks of direct comparisons between the 2 radiotracers in staging setting [17]. "
3) Ceci F, Rovera G, Iorio GC, Guarneri A, Chiofalo V, Passera R, Oderda M, Dall'Armellina S, Liberini V, Grimaldi S, Bellò M, Gontero P, Ricardi U, Deandreis D. Event-free survival after 68 Ga-PSMA-11 PET/CT in recurrent hormone-sensitive prostate cancer (HSPC) patients eligible for salvage therapy. Eur J Nucl Med Mol Imaging. 2022 Jul;49(9):3257-3268. doi: 10.1007/s00259-022-05741-9. Epub 2022 Feb 26. PMID: 35217883; PMCID: PMC9250462.
4) Rovera G, Grimaldi S, Dall'Armellina S, Passera R, Oderda M, Iorio GC, Guarneri A, Gontero P, Ricardi U, Deandreis D. Predictors of Bone Metastases at 68Ga-PSMA-11 PET/CT in Hormone-Sensitive Prostate Cancer (HSPC) Patients with Early Biochemical Recurrence or Persistence. Diagnostics (Basel). 2022 May 24;12(6):1309. doi: 10.3390/diagnostics12061309. PMID: 35741119; PMCID: PMC9221902.
These refs could have been inserted and discussed within Discussion as well, but that Section is already fluent and can remain unchanged. Thus, better to improve the introduction.
Author Response
Thank you for your appreciating the topic of our work and for your valuable suggestion for improving it.
Following your recommendations, we added the references and rephrased the part of introduction indicated.